# Student Thoughts on Virtual Reality in Higher Education—A Survey Questionnaire

**Igor Cicek [1], Andrija Bernik [2],*  and Igor Tomicic [3]**

1   ProtoPixel, Pere IV 78 2do 1ra, 08005 Barcelona, Spain; icicek1@gmail.com
2   Department of Multimedia, University North, 42000 Varazdin, Croatia
3   Faculty of Organization and Informatics, University of Zagreb, 10000 Zagreb, Croatia; itomicic@foi.unizg.hr
*   Correspondence: abernik@unin.hr

**Abstract:** This paper explores the benefits of using Virtual Reality (VR) technologies in higher education. The theoretical part investigates the classical education system and its features in order to compare advantages of using VR systems in education. VR technologies and its current state in industry and in education were explored in addition to which branches of higher education use these systems. A survey was conducted through an online questionnaire where respondents (N = 55) gave their opinion on VR and the implementation of VR technologies in education. Three hypotheses related to the use of VR technology, student interest, and learning outcomes as well as the effectiveness, immersiveness and the effect of VR systems on the users were tested through 27 questions.

**Keywords:** virtual reality; higher education; technology; immersion

## 1. Introduction

The use of digital devices for learning and education purposes is increasingly widespread. This is particularly noticeable in the period from 1997 to 2006, when networked computers were extensively used for shared learning, and in the period from 2007 to 2016, when so-called online digital learning became widespread. During these two periods, people were questioning the potential to leverage new technologies such as virtual learning environments and mobile devices. Although, traditional education has emphasized the teacher rather than the student, this approach has shown some major flaws, and it is not relevant by today's standards [1]. As Colin and O'Brien stated [2], students are instructed to do their own peer review, to experiment on their own, and to try to connect their findings with known knowledge rather than passively accept what teachers offers to them. Technology has a major role in this trend, especially gamification [3–6], augmented reality (AR) and virtual reality (VR) entertainment and educational mobile devices [7,8]. Lately, VR technologies have actively been used in education, teaching, and training in various implementation domains [9]. Even though VR is not new, the development of immersive technologies in the last ten years in terms of visualization and interaction has made VR more appealing to scientists. The latest VR screens, such as HTC Vive or Oculus Rift, allow users to experience a high degree of immersion. The term "immersion" describes a user's participation in a virtual environment during which their real-time awareness of time often becomes incoherent. It is projected that the Head Mounted Display (HMD) market reach an estimated USD 25 billion by 2022, growing at an annual rate of 39.52% between 2019 and 2025. Therefore, this is the perfect time to explore immersive VR, primarily due to the increased capabilities of VR technology, as well as increasingly affordable prices [9].

There are 3300 higher education institutions in the European Union. Compared to the US system, the European system is much more complex, since it is primarily organized at national and regional levels, each with its own legal requirements, cultural and historical frameworks, and different languages [10]. In order to allow for a uniform

approach, three key dimensions from the literature can be singled out: "stratification", "specificity/orientation", and "standardization". These key dimensions are simplifications of the design multiplicity of education systems and, are primarily used to imply specific characteristics of a wider range of systems [11]. The terms virtual reality (VR) and virtual environment (VE) are used interchangeably in the computing community, and they are the most popular terms to describe virtual reality technology; however, there are many other terms used to describe virtual reality: synthetic experience, virtual worlds, artificial worlds, and artificial reality [12].

All of these terms denote the same type of technology: interactive real-time graphics with three-dimensional models combined with stereoscopic display technology, which allows users to immerse themselves in the virtual world and directly manipulate it [13]. It is about an illusion of participating in a synthetic environment, not about an external observation of such an environment. VR relies on a three-dimensional stereoscopic screen that monitors head, arm, and body movements, as well as binaural audio. VR is an immersive, multi-sensory experience [14]. Virtual reality refers to an immersive, interactive, multi-sensory, three-dimensional environment generated by a computer and combined with the technology needed to build such environments [15].

The first idea of virtual reality was conceptualized in 1935, when Stanley G. Weinbaum described glasses that allow the user to watch a film from a first-person perspective; the viewer was the main protagonist, and interacted fully with the film through image, sound, smell, taste, and tactile perception. When we compare modern VR systems to Weinbaum's idea, it is impossible to ignore striking similarities between his original idea and the current state of technology [16].

The key issue in the use of new technologies in education is still the ability of educators to conceptualize how to best integrate technology into the curriculum. The concept of active VR is aligned with the goal of designing pedagogical systems that focus on networked and authentic interactive experiences with the student in their center. Networked learning places great emphasis on collaboration, determination, trust, and organization in the learning process. Examples of networked student-centered pedagogies include social constructivism, branched learning, and heutagogy [17,18]. Even though the development of VR for educational purposes is noticeable, most people still use technology to advance and strengthen traditional teaching methods using the new platform without changing the pedagogy of learning and teaching. Modern technologies, including VR, should be applied in STEM education in order to increase efficiency and interest in learning and research [19]. Probably the most well-known application of VR is the one used for military purposes, which includes combat simulators in all defense departments (air force, navy, and army) [20].

Following a survey conducted in 2018 of 25 students in the field of education on the topic "Analysis and Use of VR for Educational Purposes in the Field of History", it was concluded that the available content, with the help of VR, increased long-term acquired knowledge. Through the use of VR, participants stated that they could be much more immersed in the content due to the fact that VR was a lot more exciting and interesting than traditional educational techniques, VR increased interest in and motivation for the content, and consequently contributed to better and longer-lasting mastery of educational content [21]. The disadvantages of using virtual reality in education primarily relate to the fact that it is a relatively new type of technology that has not yet been sufficiently analyzed. Disadvantages are mostly related to costs, the time required to learn to use hardware and software, possible health and safety effects on the user, and slow integration of technology into the curriculum. As with all new technologies, each of these issues can gradually be solved by conducting additional research, as well as by general acceptance of technology outside of teaching processes [22]. There is compelling evidence that students can improve their learning process through VR systems. However, there are still a number of unresolved issues regarding the efficiency of such systems. Immersive (HMD) versus non-immersive (traditional desktop screens) VR systems, collaboration on educational VR systems, and

the level of realism in VR systems are just some of the issues that arise in debates when it comes to the implementation of VR systems for educational purposes [23,24].

In the next stage we discuss the research idea, problem, purpose and hypothesis. In addition, a brief guideline for the research is given, followed by the data analysis. Three hypotheses were put to the test and respondents reactions were analyzed. Finally, conclusions are given and an Appendix A is provided with the hypotheses related survey questions.

## 2. Research Design

**Defining the problem:** Although it is obvious that virtual reality can greatly contribute to the educational process, this type of technology is still not implemented in most areas of the education system, and in cases in which it is used, it is still considered a novelty instead of a fundamental tool for improving the educational process. The question is whether interest in this type of technology is significant enough to justify costs under the state budget and changes in the education system in order for VR technology to be used effectively in education.

**Purpose and goal of the research:** Since the use of new technologies is perceived as a step towards progress in all aspects of human life, the education system is looking for new ways to improve the use of new technologies as well. VR has experienced rapid development in the last decade and has become increasingly present in everyday life. All previous research points to a large number of advantages that VR can offer in education and a very small number of disadvantages. Based on previous research, the aim of this study is to examine the acceptability and general attitudes of people towards the introduction of VR systems in educational institutions.

**Hypothesis 1 (H1)**. *Respondents prefer to use an HMD VR over a 2D display*.

**Hypothesis 2 (H2)**. *Respondents believe that the use of VR systems would increase interest in certain teaching content*.

**Hypothesis 3 (H3)**. Respondents believe that the introduction of interactive media (in this case, VR systems) into the curriculum would improve learning outcomes.

**Respondents:** The survey was conducted on respondents who voluntarily completed the questionnaire. Fifty-five respondents, of whom 30 were men and 25 women, completed the questionnaire. Respondents had different levels of education; the largest percentage of participants (38.2%) had some form of higher education (college, university, college, etc.). The majority of respondents (52.7%) stated that they had tried to use a VR system more than once.

**Measuring instruments**: Attitudes of respondents and the acceptability of the introduction of VR systems in education were examined with the aid of the Likert scale. The statements used in the scale were selected based on a study of the literature related to the effectiveness of VR systems in education and to the immersiveness and impact of VR systems on the user. The questionnaire was conducted in English in order to include as large sample as possible of people with different demographic backgrounds. An example of how the questionnaire was made available to the respondents can be found in the "Attachments" of this paper.

**Procedure**: The survey was conducted in the period from 23 to 29 August 2020. The survey was created with the help of the "Google Forms" application and was distributed on various internet forums. Participants volunteered to fill out the questionnaire, which was anonymous. The questionnaire was based entirely on the opinions of the participants. At the beginning of the questionnaire, its context and purpose were briefly described. In the first part of the questionnaire, the respondents were asked general demographic questions. In its longer part, the questionnaire was based on the Likert scale. The statements were

related to the assumed hypotheses, and control questions were included. The average time required to fill out the questionnaire was six minutes.

## 3. Results

Upon completion of the analysis of individual items, taking into account control and inverse questions, which must be transformed in the same way, the calculation was made as to indicate a positive attitude towards a statement/hypothesis. Hypotheses were evaluated according to the average percentage of respondents who expressed positive attitudes with regards to the statements related to a particular hypothesis, based on mode and median. On a scale of 1 to 5, answers 4 and 5 were considered positive, answer 3 was considered neutral, and answers 1 and 2 were considered negative. In order for a hypothesis to be confirmed with certainty, the percentage of positive answers had to exceed 51%, while mode and median had to exceed the limit of 3.75.

Fifty-five respondents, of whom 30 were men and 25 women, participated in the research. Most of the participants (30) were part of the education system, 21 respondents had completed higher education, and 4 respondents had not completed any form of higher education. Forty-six respondents had the opportunity to test a VR system, and nine respondents never had an experience with a VR system. Of the 55 respondents, 25 had a VR system at home.

**Hypothesis** 1 examines whether users prefer to use an HMD VR system over a 2D display. This applies to all types of content: interactive content such as video games, use of 3D applications, and passive content such as video on demand and the like. Respondents were presented with a series of statements that question respondents' interest in certain elements of the VR system, such as the sense of passage of time (T1), the sense of presence in the virtual world (immersion) (T2, T3), and opinion on VR technology (T4, T5, T6). Every statement (T) is given in following table. A control question (T3) was posed as well. When it come to the second statement (T2), 60% of the respondents reacted positively, and 52% of them reacted positively to the control question (T3). Data analysis found that the respondents were not paying close attention or did not fully understand the control question. Due to the inconsistency of the data, statement two (T2) and statement three (T3) were not be taken into account when evaluating the hypothesis. The statement "Complete immersion in the virtual world frightens me" was an inversion, and it was necessary to inversely recalculate the mode and median accordingly, while the percentage of respondents who did not agree with this statement was taken into account as a percentage. When we excluded the above statements (T2, T3) from the evaluation, the average percentage of respondents who agreed with the statements related to Hypothesis 1 was 68.25%, while the average mode value was 4.5 and the average median value was 4.00. According to all indicators, the hypothesis "Respondents prefer to use an HMD VR over a 2D screen" could be confirmed, but it is necessary to take into account the inconsistency of answers to certain statements; after the elimination of inconsistent statements, this hypothesis was confirmed on a small number of statements. See Table 1 for more details.

**Hypothesis** 2 examines whether respondents believe that the use of VR systems would increase interest in certain teaching content. The participants expressed their personal opinion on the interest in using VR in classrooms and outside of classrooms for educational purposes (T8, T12, T13, T14, T16), the importance of the social aspect in the educational process (T9, T11), and the improvement and better understanding of teaching content (T7, T10, T15). See Table 2 for more details regarding statements (T). From the available data, it is possible to conclude that there is a great interest when it comes to the use new technologies in this case, VR for teaching purposes in order to increase interest. However, the social interaction provided by educational institutions was still extremely important for the respondents, and the "learning from home" model through Video on Demand (VoD) content was not desirable, while social interaction in a virtual environment was acceptable. Respondents also believed that the use of VR in education would not distract a student from their teacher's content. The control question (T15) was consistent when compared

to the other answers. The average percentage of positive opinions about the statements related to this hypothesis was 66.1, while the average mode of statements was 4.3, and the median was 3.90. According to all indicators, the hypothesis "Respondents believe that the use of VR systems would increase interest in certain teaching content" was confirmed. See Table 2 for more details.

**Table 1.** Results of the questionnaire for Hypothesis 1.

| Respondents Prefer to Use HMD VR over 2D Display | | | | | | | | |
|---|---|---|---|---|---|---|---|---|
| **Statement** | **Pozitive** | | **Neutral** | | **Negative** | | **Mode** | **Median** |
| | **n** | **%** | **n** | **%** | **n** | **%** | | |
| T1: Time passes faster for me while I consume content via a VR system compared to consuming content via regular 2D displays. | 22 | 40 | 21 | 38 | 12 | 22 | 3 | 3.00 |
| T2: While I use a VR system, I feel like I am present in a virtual world. | 33 | 60 | 16 | 29 | 6 | 11 | 4 | 4.00 |
| T3: While I use a VR system, I am always aware that I'm in virtual world and that none of it is real. * | 29 | 52 | 18 | 33 | 8 | 15 | 3 ** | 2.00 ** |
| T4: With VR, I'm not limited to passively consuming information and images displayed on the screen. | 42 | 77 | 8 | 14 | 5 | 9 | 5 | 4.00 |
| T5: Complete immersion in the virtual world frightens me. * | 9 | 16 | 13 | 24 | 33 | 60 | 5 ** | 4.00 ** |
| T6: The visual stimuli provided by VR systems is fascinating to the users. | 53 | 96 | 1 | 2 | 1 | 2 | 5 | 5.00 |

* Negatively formulated statement; ** value calculated using inverse data.

**Table 2.** Results of the questionnaire for Hypothesis 2.

| Respondents Believe that Using a VR System Would Increase Interest in Certain Teacher Content | | | | | | | | |
|---|---|---|---|---|---|---|---|---|
| **Statement** | **Pozitive** | | **Neutral** | | **Negative** | | **Mode** | **Median** |
| | **n** | **%** | **n** | **%** | **n** | **%** | | |
| T8: It's difficult for me to understand abstract contents and concepts (e.g., energy transfer and similar) without a visual representation of the same. | 34 | 62 | 11 | 20 | 10 | 28 | 5 | 4.00 |
| T9: I think that my interest in courses and educational content would be higher if interactive content and VR systems were used. | 38 | 69 | 9 | 17 | 8 | 14 | 5 | 4.00 |
| T10: The group's shared experiences in a shared environment are important. | 41 | 74 | 10 | 18 | 4 | 8 | 4 | 4.00 |
| T11: Stimulation of multiple senses leads to a better understanding of educational content. | 46 | 83 | 6 | 11 | 3 | 6 | 5 | 4.00 |
| T12: Interaction with the real people in the real world, whether they are lecturers or students, is necessary. * | 37 | 67 | 4 | 7 | 14 | 26 | 2 ** | 2.00 ** |
| T13: While using VR systems, students can actively learn and participate, instead of passively looking at 2D displays. | 41 | 74 | 7 | 13 | 7 | 13 | 5 | 4.00 |
| T14: Being able to see and experience the various locations around the world within the classroom provided by VR can inspire and intrigue students. | 51 | 93 | 4 | 7 | 0 | 0 | 5 | 5.00 |
| T15: Introducing virtual reality into the classrooms turns learning into entertainment. | 39 | 71 | 9 | 16 | 2 | 4 | 4 | 4.00 |
| T16: Using a VR system would distract students from the educational content. * | 16 | 29 | 11 | 20 | 28 | 51 | 4 ** | 4.00 ** |
| T17: Due to the simulation and experience provided by VR, students will continue to explore and research the educational content. | 32 | 58 | 15 | 27 | 8 | 25 | 4 | 4.00 |

* Negatively formulated statement; ** value calculated using inverse data.

**Hypothesis** 3 examines the opinion of the respondents with regard to the belief that the introduction of VR in education and in the curriculum would improve learning outcomes. Many studies to date point to potential advances in learning outcomes when

using VR systems. Through statements related to this hypothesis, the respondents' opinion on this topic was examined. Respondents were offered statements expressing the efficiency of the learning process (T17, T18), statements related to the education system, the way information is transmitted, and the way students are evaluated (T19, T20, T21, T22, T23), and statements related to the integration of VR into education (T24, T25, T26, T27). See Table 3 for more details regarding statements (T), The T22 control question, which was based on the T21 statement, indicated that the respondents carefully read and understood the statements. From the presented data, it is possible to conclude that most people agreed with the statement that the current evaluation system (e.g., exams) does not reflect real knowledge, but that is instead necessary to find an alternative tailored to each individual. When it came to the statement that a professor should be the main source of information and interaction in the classrooms (T19), as opposed to the statement that the majority of interaction should take place among students, with a professor only servinge as a "guide" (T20), the opinions of the respondents were divided. When it came to the first statement (T19), 56% of the respondents had positive opinions, while 44% of them had positive opinions with regard to the second statement (T20). The average percentage of people who reacted positively to the statements related to Hypothesis 3 was 63.27%, the average mode was 3.9, and the median was 4.00. Based on the presented data, the hypothesis "Respondents believe that the introduction of interactive media (in this case, VR systems) in the curriculum would improve learning outcomes" could be confirmed. See Table 3 for more details.

**Table 3.** Results of the questionnaire for Hypothesis 3.

| Respondents Believe that the Introduction of Interactive Media (in this Case, VR Systems) into the Curriculum Would Improve Learning Outcomes | | | | | | | | |
|---|---|---|---|---|---|---|---|---|
| **Statement** | **Pozitive** | | **Neutral** | | **Negative** | | **Mode** | **Median** |
| | **n** | **%** | **n** | **%** | **n** | **%** | | |
| T17: People learn better through interaction. | 50 | 91 | 4 | 7 | 1 | 2 | 5 | 5.00 |
| T18: Through the learning process, it's necessary to apply theoretical knowledge to practical examples in order to master a new skill. | 49 | 89 | 5 | 9 | 1 | 2 | 5 | 5.00 |
| T19: In the classrooms, there should be mostly interaction between students (the professor only serves as a "guide" to the conversation). | 31 | 56 | 13 | 24 | 11 | 20 | 4 | 4.00 |
| T20: In classrooms, the professor should lead the keynote, i.e., the professor is the main source of information and interaction. * | 24 | 44 | 12 | 22 | 19 | 34 | 2 ** | 3.00 ** |
| T21: The classical evaluation system in education (e.g., exams) does not reflect the real knowledge of the respondents. | 38 | 69 | 11 | 6 | 11 | 20 | 4 | 4.00 |
| T22: The classical evaluation system in education (e.g., exams) reflects the real knowledge of the respondents. * | 9 | 16 | 9 | 16 | 37 | 68 | 4 ** | 4.00 ** |
| T23: Evaluation tailored to the individual, where certain parameters of the respondents are monitored with the help of VR systems represents a better evaluation system. | 33 | 60 | 15 | 27 | 7 | 13 | 4 | 4.00 |
| T24: Virtual environment models teach and train with the same efficiency as reality | 16 | 29 | 19 | 35 | 20 | 36 | 3 | 3.00 |
| T25: Unlike VR, which can provide an interactive experience, classical learning boils down to providing facts only. | 29 | 53 | 11 | 20 | 15 | 27 | 4 | 4.00 |
| T26: Virtual reality develops students' creativity. | 36 | 65 | 11 | 20 | 8 | 15 | 4 | 4.00 |
| T27: With the help of virtual reality, a student can learn how to react in certain (unknown, dangerous) situations. | 45 | 82 | 8 | 14 | 2 | 4 | 4 | 4.00 |

* Negatively formulated statement; ** value calculated using inverse data

## 4. Discussion

VR technologies have been around for some time now, especially in the field of military training and simulators. But the question remains whether it has a purpose other than

entertainment whether it is convenient for education usage. We conducted a relatively simple experiment involving 55 respondents, of which more than 50% used VR before in some form. Three hypothesis were formed as well as a survey questionnaire that supported them. After the experiment it was found that respondents do prefer to use an HMD VR over a 2D screen, but this statement has some drawbacks since "use" is not adequately defined. It is not the same to use VR for simulation, playing casual games, or searching something particular on the Internet. In this case, finding something specific via VR is something to investigate in the future. Will the respondents find that something and in what time, with regard to their control group, will they do the same via 2D monitors? Next we analyzed questions regarding Hypothesis 2, which stated "Respondents believe that the use of VR systems would increase interest in certain teaching content". We could argue that this statement was valid and that overall it was accepted by the respondents; however, there is a problem with the concept of educational content. Those who have tried to create some kind of 3D art/3D model/3D scene know that VR content on that scale (many courses per year, many subjects and examples) is very difficult task to create. However, as an idea, it is something to look forward to. Finally, the usage of VR technology in education could improve learning outcomes because of the greater student involvement and greater motivation. However, subject matter and the intended use of VR applications are important issues. More experiments need to be done on this subject with the regard to the open questions mentioned here and in similar research papers.

## 5. Conclusions

The research was based on the assumption that respondents prefer to use an HMD VR over a 2D screen and that the use of a VR system as an interactive medium would increase interest in certain teaching content, thus improving learning outcomes. The review of previous research involving VR in education indicates that the advantages offered by this interactive medium outweigh its disadvantages. The conducted research, based on the obtained results, is another part of this series and confirms the advantages of using VR systems, especially since the research encompasses mainly young individuals who are still in the education system, as well as individuals who have left the system and already have experience and opinions on the current state of the education system. However, even though the results suggest that there is interest in using new technologies in this case, VR systems it is necessary to take into account that social interaction is still important for respondents, whether in an educational institution or in a virtual environment.

Even though unresolved issues regarding the introduction of this interactive medium in education systems still need to be addressed, all of the data that were studied suggest that the benefits of this kind of technology are great; this conclusively confirms the fact that the methods for introducing VR systems into education need to be seriously considered in order to provide students with an alternative to traditional education, which would consequently improve the learning process.

**Author Contributions:** Conceptualization, I.C.; formal analysis, I.C.; methodology, A.B.; resources, I.T.; supervision, A.B.; visualization, I.T.; writing—original draft, I.C.; writing—review and editing, A.B. All authors have read and agreed to the published version of the manuscript.

**Funding:** This research received no external funding.

**Data Availability Statement:** All data from this research is available via request: abernik@unin.hr.

**Conflicts of Interest:** The authors declare no conflict of interest.

## Appendix A

*Appendix A.1 VR Systems in Education*

The survey is anonymous and opinion-based. Data will be solely used for the purposes of making the final thesis on the topic: VR systems in education.

The first part of the survey includes general questions about the examinees used to determine data blocks. The second part includes questions about the viability of VR systems in education.

In the context of this survey, VR refers to the head-mounted display type of VR system (e.g., Oculus Rift, HTC Vive, PlayStation VR, Samsung Gear VR, Google Daydream, etc.).

*Appendix A.2 General Information about Examinee*

1. Gender?

    Male
    Female

2. Did you have the opportunity to try any type of VR system (mobile, desktop, etc.)?

    No
    Yes, once
    Yes, more than once

3. Do you have any type of VR system (mobile, desktop, etc.) at home?

    Yes
    No

4. Select what is true for you:

    I attend elementary school
    I attend high school
    I attend any form of higher education (college, university, etc.)
    I graduated any form of higher education (college, university, etc.)
    I didn't finish any form of higher education (college, university, etc.)

*Appendix A.3 Hypothesis Based Questions*

In this part of the survey, questions are directly connected to the hypotheses to determine the viability of VR systems in education. All questions are on a scale of 1 to 5, where 1 is completely disagree and 5 completely agree with the statement (see Table A2).

**Table A1.** Hypotheses related survey questions.

| | | |
|---|---|---|
| 1. | Interaction with the real people in the real world, whether they are lecturers or students, is necessary. | 1–2–3–4–5 |
| 2. | The visual stimuli provided by VR systems is fascinating to the users. | 1–2–3–4–5 |
| 3. | Stimulation of multiple senses lead to a better understanding of educational content (positive stimulation to the senses consequently lead to more impactful experiences and understanding of educational content). | 1–2–3–4–5 |
| 4. | The classical evaluation system in education (e.g., exams) does not reflect the real knowledge of the respondents. | 1–2–3–4–5 |
| 5. | People learn better through interaction. | 1–2–3–4–5 |
| 6. | Complete immersion in the virtual world frightens me. | 1–2–3–4–5 |
| 7. | Time passes faster for me while I consume content via VR system compared to consuming content via regular 2D displays. | 1–2–3–4–5 |
| 8. | Introducing virtual reality into the classrooms turns learning into entertainment. | 1–2–3–4–5 |
| 9. | Through the learning process, it's necessary to apply theoretical knowledge to practical examples in order to master a new skill. | 1–2–3–4–5 |
| 10. | Due to the simulation and experience provided by VR, students will continue to explore and research the educational content. | 1–2–3–4–5 |
| 11. | Virtual reality develops students' creativity. | 1–2–3–4–5 |

**Table A2.** Hypotheses related survey questions.

| | | |
|---|---|---|
| 12. | Unlike VR, which can provide an interactive experience, classical learning boils down to providing facts only. | 1–2–3–4–5 |
| 13. | While I use a VR system, I am always aware that I'm in a virtual world and that none of it is real. | 1–2–3–4–5 |
| 14. | The group's shared experiences in a shared environment are important. | 1–2–3–4–5 |
| 15. | The classical evaluation system in education (e.g., exams) reflects the real knowledge of the respondents. | 1–2–3–4–5 |
| 16. | With the help of virtual reality, a student can learn how to react in certain (unknown, dangerous) situations. | 1–2–3–4–5 |
| 17. | With VR, I'm not limited to passively consuming information and images displayed on the screen. | 1–2–3–4–5 |
| 18. | Being able to see and experience the various locations around the world within the classroom provided by VR can inspire and intrigue students. | 1–2–3–4–5 |
| 19. | Virtual environment models teach and train with the same efficiency as reality. | 1–2–3–4–5 |
| 20. | While I use a VR system, I feel like I am present in a virtual world. | 1–2–3–4–5 |
| 21. | Using a VR system would distract students from the educational content. | 1–2–3–4–5 |
| 22. | In the classrooms, there should be mostly interaction between students (the professor only serves as a "guide" to the conversation). | 1–2–3–4–5 |
| 23. | It's difficult for me to understand abstract contents and concepts (e.g., energy transfer, and similar) without a visual representation of the same. | 1–2–3–4–5 |
| 24. | Evaluation tailored to the individual, where certain parameters of the respondents are monitored with the help of a VR system, represents a better evaluation system. | 1–2–3–4–5 |
| 25. | I think that my interest in courses and educational content would be higher if interactive content and VR systems were used. | 1–2–3–4–5 |
| 26. | In classrooms, the professor should lead the keynote, i.e., the professor is the main source of information and interaction. | 1–2–3–4–5 |
| 27. | While using VR systems, students can actively learn and participate instead of passively looking at 2D displays. | 1–2–3–4–5 |

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
