# Peer review of "Student Thoughts on Virtual Reality in Higher Education—A Survey Questionnaire"

_information, doi:10.3390/info12040151_

Round 1

Reviewer 1 Report

Instead of writing a lengthy text In prose, I provide a detailed list of comments and recommendations:
- Considering a journal article, this manuscript is surprisingly short. Brevity is not an issue, but not even seven pages of text rather hints to work-in-progress. This impression is supported by the very short reference list.
- The article needs some language polishing.
- The introduction is well done but would profit from additional citations.
- The introduction should conclude with giving an overview of the paper.
- Section 2 and 3 should be integrated into a Research Design section. Justification of the method lacks references to the literature. Also, the study design is only shallowly described.
- The presentation of the results is fine.
- A discussion is missing. For a journal paper, I would expect a profound, elaborated discussion, which is trying to generalize findings, relating them to the literature, and giving an outlook.
- If is good to have the appendix.

In conclusion, this is a nice manuscript, but it does not appear to be ready for journal publication. It either should be sent in similar form to a conference and be extended later for journal publication or be extended directly. Especially the discussion is needed to ensure it can contribute to the emerging theory in the understanding of VR in higher education. I, thus, recommend "reject" out of the possible options, but I really mean "revise and resubmit" because the article does not need a "major revision" but really an extension.

Author Response

We have made some improvements which have both Reviewers stated:
- Introduction part has 8 new citations
- Introduction has brief overview of the whole paper
- Section 2 and 3 are integrated into a Research Design
- Discusion is added

Some things were left as is such as formatting since everything is formatted as Template suggested.
Anyway, the updated version is here.

Kind regards and thank you!
Andrija

Reviewer 2 Report

his article suggests a current and attractive topic for the academy. The research is timely and worthwhile. The research problem is clearly defined. The authors provide fresh insight into the field.

I hope you find the following observations helpful:

Structure: Articles should be reformatted according to a standard structure, which is set out in the instructions for authors of the journal (sections are Introduction, Materials and Methods, Results, and Discussions, Conclusion). See new template.

Results: Perhaps it is better to visualize in more charts based on statistical methods of calculation. In my opinion, it may be better to provide the results of testing these methods (if any) in the Results section.

Authors should discuss the results and how they can be interpreted from the perspective of previously published studies.

Possibly you will need to update your reference with a published article: 

  1. R. Korzh, A. Peleschyshyn, Yu. Syerov, S. Fedushko, Principles of University’s Information Image Protection from Aggression. Proceedings of the XIth International Scientific and Technical Conference (CSIT 2016). Lviv, Lviv Polytechnic Publishing House, 2016. pp. 77-79.

Overall, I find the paper adequate but it can be improved by addressing the aforementioned issues. Especially the problem of the paper structure and lack of practical results or its representation.

For the rest, congratulations to the researchers. They have carried out a magnificent study.

Author Response

(The authors gave the same response as above.)

Round 2

Reviewer 1 Report

Dear authors,

thank you for your revision.

I still think that the paper is very short for what the title implies, but it is self-contained now, and it is sound. Thus, I think it can be accepted, although changing the title to be somewhat less general might be an option.

I suggest, however, to have a minor revision before acceptance in which the language is polished.